# SIMPLEMIX: Frustratingly Simple Mixing of Off- and On-policy Data in Language Model Preference Learning

**Tianjian Li** [1]   **Daniel Khashabi** [1]

## Abstract

Aligning language models with human preferences relies on pairwise preference datasets. While some studies suggest that on-policy data consistently outperforms *off*-policy data for preference learning, others indicate that the advantages of *on*-policy data may be task-dependent, highlighting the need for a systematic exploration of their interplay.

In this work, we show that on-policy and off-policy data offer *complementary* strengths in preference optimization: on-policy data is particularly effective for reasoning tasks like math and coding, while off-policy data performs better on open-ended tasks such as creative writing and making personal recommendations. Guided by these findings, we introduce SIMPLEMIX, an approach to combine the complementary strengths of on-policy and off-policy preference learning by simply mixing these two data sources. Our empirical results across diverse tasks and benchmarks demonstrate that SIMPLEMIX substantially improves language model alignment. Specifically, SIMPLEMIX improves upon on-policy DPO and off-policy DPO by an average of 6.03% on Alpaca Eval 2.0. Moreover, it outperforms prior approaches that are much more complex in combining on- and off-policy data, such as HyPO and DPO-Mix-P, by an average of 3.05%.

## 1. Introduction

Alignment of Language Models (LMs) has bestowed them with the ability to learn better from demonstrations (Brown et al., 2020), extrapolate from reasoning chains (Wei et al.,

[1]Center for Language and Speech Processing, Johns Hopkins University, Baltimore, US. Correspondence to: Tianjian Li <tli104@jhu.edu>.

*Proceedings of the $42^{nd}$ International Conference on Machine Learning*, Vancouver, Canada. PMLR 267, 2025. Copyright 2025 by the author(s).

2022), and produce responses aligned with human values (Ouyang et al., 2022).

The ongoing debate (Ivison et al., 2024; Tajwar et al., 2024; Tang et al., 2024; Xu et al., 2024c) in the literature that compares alignment with on-policy data (data is sampled from the LM to be aligned) with off-policy data (data is sampled not from the LM to be aligned). The debate arrives at mismatched conclusions, with some works reporting on-policy outperforms off-policy (Tajwar et al., 2024; Xu et al., 2024c) while others reporting that the gains from on-policy training are minimal, and sometimes even underperform off-policy training (Ivison et al., 2024; Ahmadian et al., 2024; Lambert et al., 2024a). Existing work (Ivison et al., 2024; Xu et al., 2024c) that compares on- and off-policy data does not control using the same algorithm under different data sources, and does not investigate the impact of task types (Tang et al., 2024).

To address these weaknesses, we vary *only* the data source and break down the performances by different task types to answer the following research questions:

($Q_1$) *Under what circumstances on- vs off-policy data offer different strengths?*

($Q_2$) *Can we leverage their complementarity for more efficient alignment?*

With regards to ($Q_1$), we observe that on-policy data and off-policy data offer complementary strengths in preference learning of LMs: *On-policy* data is most effective in tasks that tend to have objective answers and require reasoning skills, e.g., math and coding. *Off-policy* data is effective in more open-ended tasks, e.g., creative writing and making personal recommendations, where it is common to have multiple differing preferred responses (§3).

To address ($Q_2$), we propose SIMPLEMIX: to mix on- and off-policy data for preference learning. Our approach consistently improves upon only using on- or off-policy data solely in data-constrained setups where the same amount of responses is used for each method. Furthermore, we show that SIMPLEMIX is able to outperform prior approaches that mix on- and off-policy data in more complex ways (Song et al., 2024b; Shi et al., 2024), showcasing the effectiveness

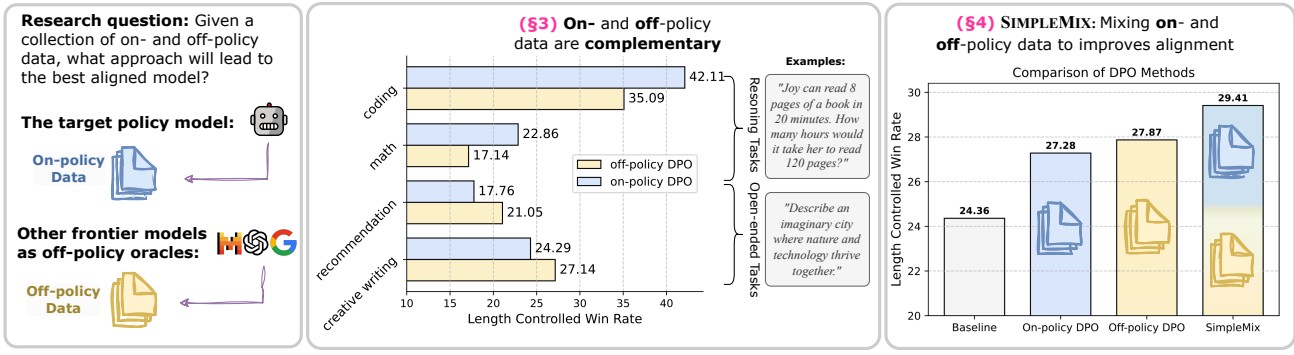

*Figure 1.* **Left panel:** Our work studies data origin in preference optimization of LMs. **Middle panel:** We show that on-policy data and off-policy data are complementary: on-policy data mostly improves the model's performance on reasoning tasks that are objectively correct or incorrect (e.g., Math and Coding) while off-policy data improves on sub-tasks where humans might disagree with each other (e.g., creative writing and personal recommendation) (§3). **Right panel:** Our proposed method SIMPLEMIX mixes on and off-policy data, outperforming solely using either on or off-policy data. (§4)

of our method despite its simplicity (§4).

Figure 1 illustrates our work: The complementary nature of on- and off-policy data in LM preference learning: on-policy DPO (Rafailov et al., 2024) outperforms off-policy DPO in reasoning tasks, e.g., solving a math problem, while underperforms in open-ended tasks, e.g., describing an imaginary city(§3); SIMPLEMIX: Mixing on- and off-policy data in DPO can outperform DPO with a single data source (§4). SIMPLEMIX achieves an average improvement of 6.03 over on-policy and off-policy methods on Alpaca Eval 2.0 and outperforms existing hybrid approaches by 3.05 while maintaining simplicity.

To sum up, our contribution is two-fold:

(1) We show that on- and off-policy data in preference learning are complementary: on-policy data excels at tasks requiring reasoning and objective verification, such as math and coding, whereas off-policy data is more effective for open-ended tasks, such as creative writing and personal recommendations.

(2) We demonstrate that a good balance between on- and off-policy data consistently outperforms approaches relying solely on either data source.

## 2. Preliminaries and Notations

We consider post-training of LMs. Given a user prompt $x$ in natural language, the language model $\pi(\cdot|x)$ takes the prompt as input and outputs a probability distribution over natural language responses $y$. The Supervised Fine-Tuning (SFT) stage of LM updates a pre-trained LM $\pi_{\text{base}}$ on a dataset with (prompt, response) pairs: $\mathcal{D}_{\text{SFT}} = \{(x_1, y_1), \ldots, (x_N, y_N)\}$ with the following maximum likelihood objective: $\pi_{\text{SFT}} \in \arg\max_\pi \prod_{i=1}^N \pi(y_i \mid x_i)$.

**Preference Optimization (PO)** While SFT is a step forward in aligning LMs to follow human instructions, the resulting model is not aligned to human preferences. For example, humans often prefer structured, concise yet complete responses. To align $\pi_{\text{SFT}}$ to human preferences, we perform Preference Optimization (PO) on top of $\pi_{\text{SFT}}$ with a pairwise preference distribution: $\mathcal{D} = \{(x, y_w, y_l)...\}$, where there are two responses $(y_w, y_l)$ for a single prompt. $y_w$ is preferred by human annotators (usually referred to as the "winning" or "chosen" response), and $y_l$ is dispreferred (usually referred to as the "losing" or "rejected" response). In essence, PO aims to steer $\pi_{\text{SFT}}$ towards the winning responses that are preferred by humans. Formally, the objective is defined as (Rafailov et al., 2024):

$$\mathbb{E}_{(x,y_w,y_l)\sim\mathcal{D}}\left[\log\sigma\left(\beta\log\frac{\pi_\theta(y_w \mid x)}{\pi_{\text{SFT}}(y_w \mid x)}\right.\right.$$
$$\left.\left.-\beta\log\frac{\pi_\theta(y_l \mid x)}{\pi_{\text{SFT}}(y_l \mid x)}\right)\right], \quad (1)$$

where $\sigma$ is a non-decreasing function. Maximizing (1) should maximize the increase in log-likelihood of $y_w$ and the decrease of $y_l$. Ideally, the model learns to generalize characteristics that make $y_w$ preferred. For example, if $y_w$ are constructed that are factually correct and $y_l$ contains hallucinations, the hope is that by steering the policy towards a few factually correct responses, the policy generalizes so that it assigns high probability to *all* responses that are factually correct. We differentiate between on- and off-policy PO depending on $\mathcal{D}$: from which $y_w$ and $y_l$ are sampled.

**On-policy PO** Assuming that the LM $\pi$ is parameterized by $\theta$. If the responses $y$ are sampled from a policy that is derived from either $\pi_{\text{SFT}}$ or $\pi_\theta$, we describe $y$ as **on-**

**policy data**: where $\mathcal{D} = \mathcal{D}_{\text{on}} = \{(x, y) \mid y \sim \pi_\theta \text{ or } \pi_{\text{SFT}}\}$.[1] Notable examples of on-policy PO include Proximal Policy Optimization (PPO) (Schulman et al., 2017) that optimizes a mathematically equivalent objective as equation (1) using on-policy generations $y \sim \pi_\theta$, and "on-policy DPO" (Guo et al., 2024) that uses a stronger language model as a judge to label pairs of on-policy generations and optimizes equation (1) with $y_w, y_l \sim \pi_\theta$.

**Off-policy PO**  Recall that on-policy data are sampled from a policy that is derived from either $\pi_{\text{SFT}}$ or $\pi_\theta$, we describe data that are not on-policy data as off-policy data and PO performed on top of an off-policy dataset $\mathcal{D} = \mathcal{D}_{\text{off}}$ Usually, off-policy data are sampled from a collection of open-sourced or proprietary models (Cui et al., 2024; Wang et al., 2024; Bai et al., 2022a). A notable example of off-policy PO is Direct Preference Optimization (Rafailov et al., 2024) (DPO) which directly optimizes equation (1) with the responses sampled from an off-policy dataset.

# 3. On- vs. Off-policy Data are Complementary

In this section, we vary the data source in DPO, whether $\mathcal{D} = \mathcal{D}_{\text{on}}$ or $\mathcal{D} = \mathcal{D}_{\text{off}}$ in equation 1 to answer ($Q_1$). Our setup ensures a consistent setup by only varying the data source while *fixing* the alignment algorithm, in contrast to prior work that does not control the alignment algorithm by comparing on- vs. off-policy data.

**Our Hypothesis**  We found that in Ivison et al. (2024), PPO shows most gains on GSM8k (Cobbe et al., 2021), a dataset consisting of grade-school math questions, while being similar in performance to DPO on general tasks such as MMLU (Hendrycks et al., 2020). In Tang et al. (2024), the performance gap between on- and off-policy data is smallest on chatbot arena side by side (Chiang et al., 2024a), which also attests to the policy's performance on general human queries. We thus make the following hypothesis:

> **Hypothesis:** On-policy data mostly helps in tasks that are objectively correct or not (e.g., math), whereas the difference between on- and off-policy data on open-ended tasks (e.g., creative writing) is minimal.

We describe our experiment setup in §3.1 and report our results that validate our hypothesis in §3.2.

---

[1] Our definition of "on-policy" coincides with Lambert et al. (2024a). Note that minor differences exist between our definition of on- vs. off-policy and the traditional Reinforcement Learning literature, which treats responses (actions) that are not sampled from $\pi_\theta$ as off-policy (Song et al., 2024b; Xie et al., 2024; Xiong et al., 2024).

## 3.1. Experiment Setup

**Models and Dataset**  We use two language models as $\pi_{\text{SFT}}$: `meta-llama-3.1-8B-Instruct` (Dubey et al., 2024) and `Llama-3.1-Tulu-3-8B-SFT` (Lambert et al., 2024a). We experiment on the Ultra-Feedback dataset (Cui et al., 2024) that consists of 60k prompts. For each prompt, two responses were collected from a model pool. We treat the two responses as **off-policy** data. For each prompt, we sample $N$ responses[2] as **on-policy** data. We use the best reward model `Skywork/Skywork-Reward-Gemma-2-27B-v0.2` in RewardBench (Lambert et al., 2024b)[3] as a proxy of human judgment to score the pair of responses. We refer to this model as the *oracle* reward model. The response with the highest reward is labeled as $y_w$ and the response with the lowest reward is labeled as $y_l$. More detailed hyperparameters are reported in Appendix A.

**Evaluation**  To test our hypothesis, we break down the prompts in Alpaca Eval 2.0 (Li et al., 2023) into different categories. A detailed breakdown of prompt category and distribution can be found in Appendix B. We select four representative tasks: two are objective tasks that can be verified: **mathematical reasoning or calculation (Math)** and **Coding**, and the other two tasks are tasks that require modeling human open-ended preference: **Creative writing** and **making personal recommendations (Recommendation)**. We report the length controlled win rate (Dubois et al., 2024) of each selected task for models trained with on- and off-policy DPO.

## 3.2. Results

**When does on-policy sampling help?**  Figure 2 shows the win rate for the 4 representative categories. We plot the detailed performance of the performance of on- and off-policy DPO on Math and Coding prompts in Alpaca Eval 2.0 at left of Figure 2, and we plot the results in Creative writing and Recommendation queries at Figure 2.

We observe that *on-policy DPO helps in objective and verifiable tasks.* Specifically, in our experiments of `meta-llama-3.1-8B-Instruct`, on-policy DPO outperforms off-policy DPO in math and coding, improving the length-controlled win rate by +5.72% and +7.02% respectively. However, on open-ended tasks: creative writing and making personal recommendations, on-policy DPO underperforms off-policy DPO by -2.85% and -3.29% respectively.

**Is the improvement about length?**  One might argue that a possible explanation is that tasks requiring longer genera-

---

[2] Sampling Parameters: temperature $\tau = 0.7$, top-p $p = 0.9$
[3] Best performing as of 2024-11-27

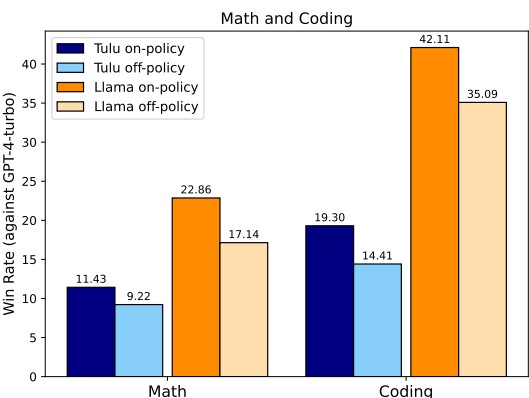
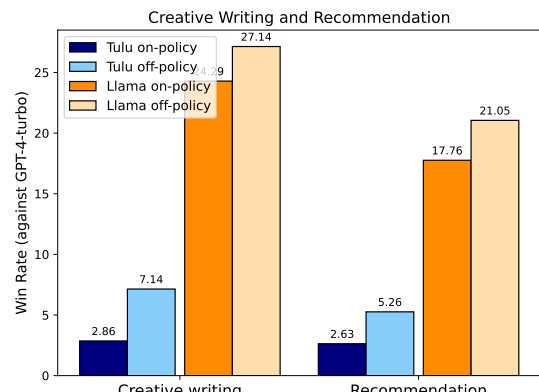

*Figure 2.* Comparison of win rates (against GPT-4-turbo) across different prompt categories in Alpaca Eval 2.0 for (left) objective tasks that have a groundtruth answer and (right) open-ended tasks where humans have individual preferences. **On-policy DPO improves performance in math and coding, while off-policy DPO demonstrates better performance in creative writing and making personal recommendations.**

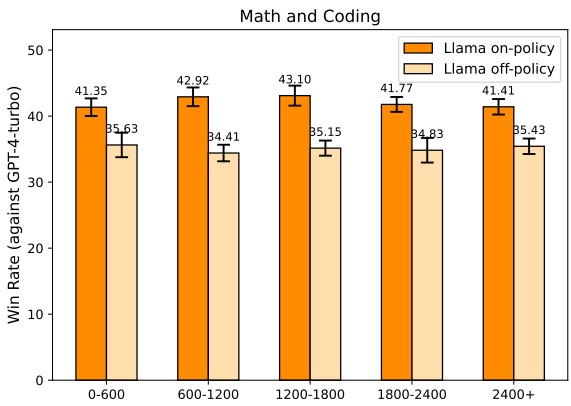
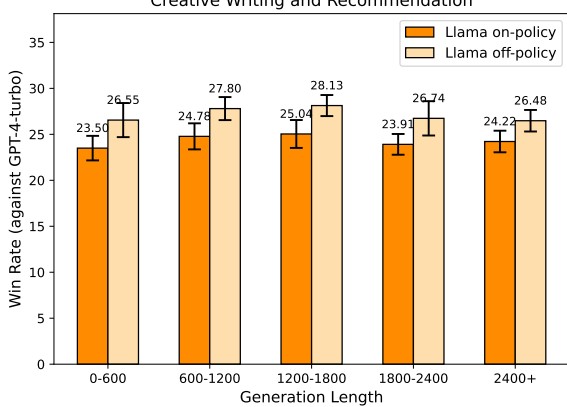

*Figure 3.* Comparison of win rates (against GPT-4-turbo) by the length of generation. On-policy training does not significantly outperform off-policy training as generation length increases in both math and coding tasks (left) and creative writing as well as recommendation tasks (right). Error bars show 95% confidence intervals from bootstrapping.

tions benefit more from on-policy data and that the improvement in math and coding can be attributed to the improvement in length. To investigate this, we plot the histogram of the length controlled win rate by the length of the generated sequences in Figure 3.

We observe that on-policy DPO does not outperform off-policy DPO as the generation length increases. This indicates that the improvements in on-policy training on math and coding cannot be solely attributed to increasing the length of the generations. The type of task is the primary factor in determining whether on-policy or off-policy training is more effective.

We conclude that on-policy DPO mostly improves on objective tasks (e.g., math and coding) while off-policy DPO excels in improving open-ended tasks (e.g. creative writing and personal recommendation) and that this distinction can-

not be explained by generation length. In the next section, we demonstrate how combining the strengths of on-policy data with the efficiency and abundance of off-policy data available on the web can lead to improved LM alignment.

## 4. SIMPLEMIX: Simply Mixing On- and Off-policy Data Improves Alignment

In this section, we show the benefits of combining on- and off-policy data in preference optimization. Specifically, we employ the data source $\mathcal{D} = \mathcal{D}_{\text{on}} \cup \mathcal{D}_{\text{off}}$ in equation 1 to answer ($Q_2$). We describe our experiment setup and baselines at §4.1, and report our main results at §4.2.

### 4.1. Setup

**SIMPLEMIX** Our method SIMPLEMIX combines on- and off-policy data, where the winning $y_w$ and losing responses

| Method Name | Sampler of $y_w, y_l$ | Loss Function |
|---|---|---|
| *Non-Hybrid Methods* | | |
| Off-policy DPO | $y_w, y_l \sim \mathcal{D}$ | $\text{DPO}(y_w, y_l)$ |
| On-policy DPO | $y_w, y_l \sim \pi_{\text{SFT}}$ | $\text{DPO}(y_w, y_l)$ |
| *Hybrid Methods* | | |
| DPO-Mix-P (Shi et al., 2024) | $y_w, y_l \sim \{\pi_\theta^{3/2} \pi_{\text{SFT}}^{-1/2}, \pi_\theta^{1/2} \pi_{\text{SFT}}^{1/2}\}$ | $\text{DPO}(y_w, y_l)$ |
| HyPO (Song et al., 2024b) | $y_w, y_l \sim \mathcal{D}, y \sim \pi_\theta$ | $\text{DPO}(y_w, y_l) - \lambda \log(y \mid x) \cdot \text{sg}^* \left( \frac{\pi_\theta(y\mid x)}{\pi_{\text{SFT}}(y\mid x)} \right)$ |
| SIMPLEMIX (Ours) | $y_w, y_l \sim \{\mathcal{D}, \pi_{\text{SFT}}\}$ | $\text{DPO}(y_w, y_l)$ |

*Table 1.* A comparison between SIMPLEMIX and other baselines we compare in this work. $^*\text{sg}()$ denotes the stop gradient operation.

$y_l$ are sampled from $\pi_{\text{SFT}}$ and the off-policy dataset $\mathcal{D}$ with **equal** probability. An in-depth study on the effect of different sampling probabilities is at §5.1.

**Baselines** We compare the following baselines with SIM­PLEMIX. A more detailed comparison of our baselines with SIMPLEMIXis in Table 1.

- **On-policy DPO**: Winning and losing response sampled from the SFT model: $y_w, y_l \sim \pi_{\text{SFT}}(\cdot \mid x)$.

- **Off-policy DPO**: Winning and losing response sampled from an off-policy dataset: $y_w, y_l \sim \mathcal{D}_{\text{off}}$.

- **HyPO** (Song et al., 2024b): Off-policy DPO with KL Regularization using on-policy data: $y_w, y_l \sim \mathcal{D}_{\text{off}}, y \sim \pi_\theta(\cdot \mid x)$. Compared to HyPO (Song et al., 2024b), SIM­PLEMIX changes the online KL regularization objective to a DPO loss.

- **DPO-Mix-P** (Shi et al., 2024): On-policy data generated with a interpolation between the current model $\pi_\theta$ and $\pi_{\text{SFT}}$. $y_w, y_l \sim \{\pi_\theta^{3/2} \pi_{\text{SFT}}^{-1/2}, \pi_\theta^{1/2} \pi_{\text{SFT}}^{1/2}\}$. Compared to DPO-Mix-P (Shi et al., 2024), SIMPLEMIX can be seen as a "hard" version of sampling from an interpolation between an off-policy LM and the SFT policy $\pi_{\text{SFT}}$ with equal weights.

**Models and Datasets** We experiment with two $\pi_{\text{SFT}}$ models `meta-llama/Llama-3.1-8B-Instruct` and `allenai/Llama-3.1-Tulu-3-8B-SFT` with DPO (Rafailov et al., 2024) and experiment on two pairwise preference datasets: Ultrafeedback (Cui et al., 2024) and Help­Steer2 (Wang et al., 2024). A detailed description of the collection of prompts and responses in both datasets can be found in Appendix C. We train our models for one epoch on the same amount of $(y_w, y_l)$ pairs for every baseline. This controls that every method has seen the same amount of data, although on-policy DPO requires more computing since it requires sampling from either $\pi_{\text{SFT}}$ or $\pi_\theta$.

**Evaluation** To evaluate the aligned policy, we follow existing literature on LM alignment and use the same 7 benchmarks in the FineWeb (Penedo et al., 2024) evaluation suite, containing Commonsense QA (CQA; Talmor et al. (2019)), Hellaswag (HS; Zellers et al. (2019)), Openbook QA (OQA; Mihaylov et al. (2018a)), PIQA (Bisk et al., 2020), Wino­grade (Sakaguchi et al., 2020), ARC-Challenge (ARC-C; Clark et al. (2018)), and MMLU (Hendrycks et al., 2020), which covers 7 knowledge and common sense based benchmarks. We also use Ifeval (Zhou et al., 2023) and Alpaca Eval 2.0 (Dubois et al., 2023; 2024) for evaluating general instruction following ability. We report the average scores of the seven tasks + Ifeval prompt level loose accuracy as **Avg. Score** and Alpaca Eval 2.0 length controlled win rate (LC). Detailed descriptions of benchmarks can be found in Appendix D.

### 4.2. Main Results

Table 2 shows the experiment results on Alpaca Eval 2.0 and averaged scores over 8 benchmarks (Avg. Score) for all of our baselines. We report the detailed breakdown of individual benchmarks in Appendix E.

We find that **SIMPLEMIX consistently yields the best performance across both models and evaluation metrics.** This indicates that the widely available source of off-policy preference data can still provide valuable signals to preference optimization. We plot the per-task performance in Alpaca Eval 2.0 of models trained with SIMPLEMIX in Figure 8 in Appendix G. SIMPLEMIX offers a balance between the performance on reasoning (math and coding) and open-ended tasks (creative writing and recommendation).

## 5. Ablations in Preference Data Curation

In this section, we investigate curation strategies in preference data curation. Specifically, we look into the impact of response diversity (5.1), the mixture ratio between on- and off-policy data (5.2), and the effect of filtering off-policy data (5.3).

| | $\pi_{\text{SFT}} =$ **Tulu-3-8B-SFT** | | $\pi_{\text{SFT}} =$ **Llama-3.1-8B-Instruct** | |
|---|---|---|---|---|
| | Alpaca Eval 2.0 LC (%) ↑ | Avg. Score ↑ | Alpaca Eval 2.0 LC (%) ↑ | Avg. Score ↑ |
| $\pi_{\text{SFT}}$ | 8.52 | 59.72 | 24.36 | 60.88 |
| *Experiments on UltraFeedback (Cui et al., 2024)* | | | | |
| Off-policy DPO | 14.23 | 60.02 | 27.87 | 61.67 |
| On-policy DPO | 15.17 | 59.70 | 27.28 | 62.09 |
| HyPO (Song et al., 2024b) | 18.11 | 60.03 | 27.91 | 62.01 |
| DPO-Mix-P (Shi et al., 2024) | 17.25 | 60.70 | 27.79 | 61.61 |
| SIMPLEMIX | **20.64** | **61.14** | **29.41** | **63.06** |
| *Experiments on HelpSteer2 (Wang et al., 2024)* | | | | |
| Off-policy DPO | 14.74 | 59.94 | 24.81 | 61.54 |
| On-policy DPO | 15.26 | 60.21 | 25.09 | 61.77 |
| HyPO (Song et al., 2024b) | 18.19 | 59.96 | 27.15 | 61.69 |
| DPO-Mix-P (Shi et al., 2024) | 17.75 | 60.04 | 27.11 | 61.49 |
| SIMPLEMIX | **21.11** | **61.22** | **29.12** | **62.12** |

*Table 2.* Performance comparison among various on and off-policy preference optimization methods on UltraFeedback (Cui et al., 2024) and HelpSteer2 (Wang et al., 2024). **Avg. Score** refers to the average score across 8 benchmarks listed in Appendix D. We found that the performance gap between on- and off-policy DPO is minimal. Incorporating off-policy data into on-policy DPO outperforms using on- or off-policy data only. Furthermore, **curating a balanced mixture of on- and off-policy data in DPO consistently yields the strongest performance in both LM-as-a-judge evaluation (Alpaca Eval 2.0) and reference-based evaluation (Avg. Score)**.

### 5.1. The Effect of Preference Data Diversity

Some works theoretically show that on-policy preference optimization benefits from more exploration (Rashidinejad & Tian, 2024; Song et al., 2024b; Anonymous, 2024a; Xiong et al., 2024). This is to say that eliciting more diverse responses would be beneficial. A possible explanation of the effect of off-policy data is that it increases the diversity of generations. Works have reported that on-policy generations usually share the same prefix (Knight et al., 2017) — leading to reduced diversity among responses. On the other hand, off-policy data, esp. Ultrafeedback, the responses are collected from a **collection** of models rather than a single model, thus improving diversity.

To validate this claim, we compare our on+off policy mixture with directly increasing the diversity of responses with:

1. **Prompting:** We iteratively prompt the language model to generate $N$ responses for diversity (Lu et al., 2024): we first prompt the language model with $x$ and obtain the first response $y_1 \sim \pi(\cdot \mid x)$, then we condition on a written prompt $z$ that asks for a diverse response for the same input $y_2 \sim \pi(\cdot \mid x, z)$. We repeat this process to obtain $N$ responses. We report the prompt $z$ in Appendix H.

2. **Large temperature:** Sampling $y_w, y_l$ with a larger sampling temperature $\tau$. We experiment with $\tau = \{1, 2, 3\}$.

We sample $N = 4$ responses from $\pi_{\text{SFT}} =$ `Llama-3.1-8B-Instruct` on the 60k prompts in Ul-

| | LC (%) ↑ | Avg. Score ↑ | Avg. Reward |
|---|---|---|---|
| Prompting | 21.41 | 58.27 | -10.75 |
| $\tau = 1.0$ | 26.94 | 60.21 | -9.44 |
| $\tau = 2.0$ | 26.77 | 59.44 | -10.56 |
| $\tau = 3.0$ | 26.18 | 58.83 | -10.77 |
| SIMPLEMIX | **29.41** | **63.06** | **-3.39** |

*Table 3.* Performance comparison between different sampling methods for eliciting diverse responses. Explicitly prompting for diversity or increasing the sampling temperature $\tau$ results in generations with lower quality, as measured by our reward model. **Low-quality generations, albeit diverse, lead to worse performance on our evaluation when performing DPO on them.**

trafeedback, again we use the oracle reward model to annotate the best and worst response as $y_w$ and $y_l$. We report the results of training $\pi_{\text{SFT}}$ on this diverse response in Table 3.

**We found that eliciting more diverse responses often comes at the cost of decreasing generation quality, thus hurting performance.** We manually inspected the generations of prompting and larger generation temperature $\tau$, and we found that most of the generation, albeit diverse, often falls short in quality compared to direct sampling. The explanation is that diversity itself is not helpful because the diversity might fall into low-reward regions, increasing diversity while reducing quality. Our findings echo (Anonymous, 2025b) who found that sampling with a larger tem-

perature leads to degraded performance on various question-answering benchmarks.

## 5.2. The Effect of On- vs. Off-policy Data Mixtures

We further investigate different data mixtures of on- and off-policy data by keeping the total amount of data the same but varying the sampling ratio between on- and off-policy data. Figure 4 shows the results of training meta-Llama-3.1-8B-Instruct and Llama-3.1-Tulu-8B-SFT on Ultrafeedback with different on- to off-policy data ratio. We observe that sampling with equal probability from both data sources (0.5 on-policy + 0.5 off-policy) outperforms other mixtures, which echos the results in Ball et al. (2023) who finds that a mixture of 0.5 off-policy data + 0.5 on-policy data generally performs the best for traditional RL tasks. Ball et al. (2023) also finds that in traditional RL tasks, a balanced 0.5 - 0.5 mixture is also the most stable. Unfortunately, based on the error bars in 4, we found no clear pattern about the variance of different mixtures.

## 5.3. The Effect of Off-policy Data Filtering

Off-policy responses are often collected with **various** open-sourced models for maximizing diversity, which often includes smaller models that are less capable. For example, responses in Ultrafeedback (Cui et al., 2024) are sampled from models ranging from the most capable GPT-4 to smaller open-sourced 7B models, often including low-quality responses. We investigate the effect of filtering out these low-quality responses when combined with on-policy data:

Specifically, we select a fraction $p$ of the entire Ultrafeedback dataset (Cui et al., 2024) according to the following heuristics:

- **Quality**: We annotate the pairs of generation with Skywork-Reward-Gemma-2-27B-v0.2. We select the top-$p$ pairs with the highest total reward.

- **On-policiness**: Existing work shows that LM learns better from on-policy data (Tajwar et al., 2024; Tang et al., 2024). We select top-$p$ of the most on-policy examples. We measure "on-policiness" by the sum of the log-probabilities of chosen and rejected responses using $\pi_{\text{SFT}}$.

- **Contrastiveness**: Existing work (Kim et al., 2024) reports that the policy learns better from highly contrastive pairs, where the quality gap between the chosen and rejected response is large. We select top-$p$ pairs where the difference between the chosen reward and the rejected reward is largest.

- **Similarity**: Existing work (Pal et al., 2024; Razin et al., 2024) points out that similar examples harms DPO performance. We take the cosine similarity of the last layer

sentence embedding between the chosen and rejected response as a proxy for "similarity" and select top-$p$ examples that are most dissimilar.

We train Llama-3.1-8B-Instruct on filtered Ultrafeedback with $p = \{0.1, 0.2, 0.3, 0.4, 0.5\}$ and report the results on Alpaca Eval 2.0 in Figure 5. We found that selecting data based on quality (orange) outperforms other criteria except when $p = \{0.5\}$.

Since we know that the quality of off-policy data has the most impact on performance, we filter the off-policy data according to its reward evaluated by our oracle reward model and mix it with the on-policy data. Specifically, we keep $p = 0.4$ fraction of Ultrafeedback and mix it with the same amount of on-policy data. We report the results in Figure 6. We found that only keeping the high-quality off-policy data, when paired with on-policy data, can further improve the performance.

## 6. Related Works

**Comparing On- and Off-Policy Alignment** The literature (Xu et al., 2024c; Ivison et al., 2024) usually compares PPO (Schulman et al., 2017) with DPO (Rafailov et al., 2024), arriving at mismatching conclusions with some works (Xu et al., 2024c) showing that PPO outperforms DPO, some works show the gap between PPO and DPO is minimal (Lambert et al., 2024a; Ivison et al., 2024). Two concurrent efforts are close to our work (Tajwar et al., 2024; Tang et al., 2024). Both studies investigate the differences between on and off-policy alignment, but Tajwar et al. (2024) conducted experiments on a controlled setting that is different from real-world LM alignment, and although Tang et al. (2024) have arrived at a conclusion that on-policy outperforms off-policy, the performance gap between on- and off-policy data in the most general task in their setting (chat arena side by side) is the smallest, which motivates us to investigate deeply. Additional related works on LM alignment is at Appendix F.

**Hybrid RL** While there has been many works that bridges on- and off-policy RL (Song et al., 2023; Wagenmaker & Pacchiano, 2023; Gu et al., 2017; Nakamoto et al., 2023; Ball et al., 2023; Lee et al., 2021; Tan et al., 2024; Song et al., 2024a), few work studies combining off-policy and on-policy data in language model alignment (Song et al., 2024b; Shi et al., 2024; Anonymous, 2024b; Bose et al., 2024), whose contributions are mostly theoretical in terms of coverage (Song et al., 2024b), convergence rates (Shi et al., 2024; Bose et al., 2024), and optimality (Xiong et al., 2024). However, these works often only conduct experiments on LLM-as-a-judge benchmarks, which can be easily hacked (Singhal et al., 2024; Wei et al., 2024; Park et al., 2024), on models that are smaller and weaker (e.g., the Pythia suite).

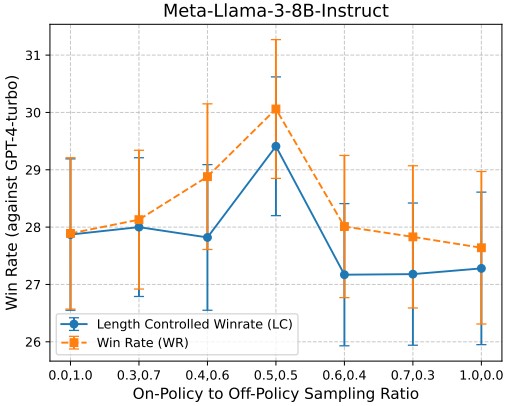
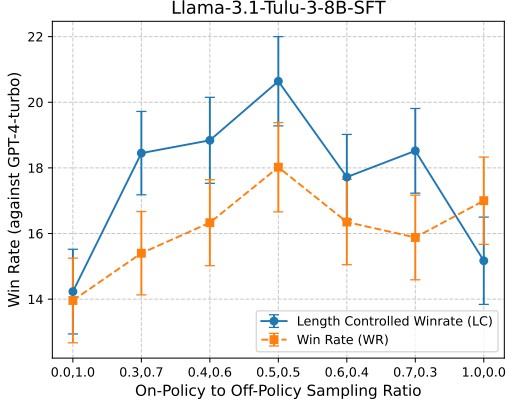

*Figure 4.* Perfomance on Alpaca Eval 2.0 for different on- to off-policy data ratio for performing DPO on top of $\pi_{\text{SFT}}$ = `Meta-LLama-3.1-8B-Instruct` (left) and `Llama-3.1-Tulu-8B-SFT` (right) on the Ultrafeedback ([Cui et al., 2024](#)) dataset. **A balanced mixture (0.5 on-policy + 0.5 off-policy) outperforms other mixtures.**

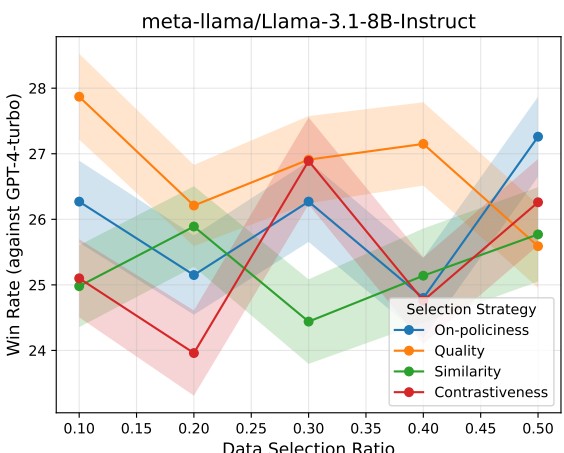
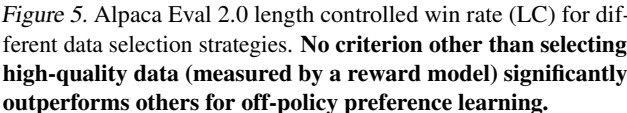

*Figure 5.* Alpaca Eval 2.0 length controlled win rate (LC) for different data selection strategies. **No criterion other than selecting high-quality data (measured by a reward model) significantly outperforms others for off-policy preference learning.**

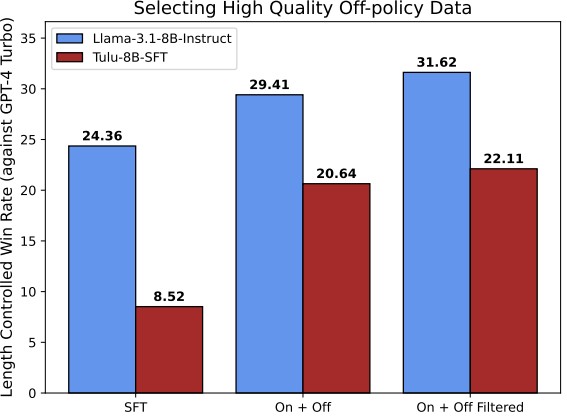

*Figure 6.* Alpaca Eval 2.0 Length Controlled Win Rate for $\pi_{\text{SFT}}$, Mixing On + Off-policy Data and Mixing in high-quality off-policy data with on-policy data in DPO. **Selecting high-quality off-policy data can improve performance using SIMPLEMIX.**

To the best of our knowledge, our work conducts the most comprehensive evaluation, including reference-based and LM-as-a-judge benchmarks, on top of state-of-the-art open-sourced models: llama-3.1-8B-Instruct ([Dubey et al., 2024](#)) and Tulu-3-8B-SFT ([Lambert et al., 2024a](#)).

## 7. Limitations and Implications

**Limiations** We acknowledge that although we exhausted our resources to perform hyperparameter searching, the combinatorically large hyperparameter space makes it challenging to draw decisive conclusions. Therefore, although we observed that the performance gap between on- and off-

policy preference optimization is minimal, it could be the case the we did not tune our model with the perfect configurations. Moreover, although we also exhausted our resources for evaluation on both reference-based and reference-free benchmarks, LLM-as-a-judge benchmarks can be hacked ([Anonymous, 2025a](#)), exhibit certain biases ([Panickssery et al., 2024](#); [Park et al., 2024](#)), and is sensitive to prompt formatting ([Zheng et al., 2023](#)). Therefore, the applicability of our conclusions is limited to the hyperparameters configurations, models, training datasets and evaluation benchmarks on which we experimented.

**Implications** Our work implies that data selection for LM alignment should be task dependent: for harder tasks where

few high-quality data is available but the answer can be easily verifiable, e.g., math, where you can perform string matching for the answer., using on-policy data, outperforms off-policy data. However, in general, for tasks where abundant data is available online, performing on-policy sampling does not outperform. Therefore, one can resort to off-policy data, saving on the cost of generating multiple responses per query. We leave the impact of fine-grained "on-policyness" of data on LM performance for future work.

## 8. Conclusion

In this paper, we show that in preference learning of LMs, *on-policy* data and *off-policy* data are complementary: on-policy data improves upon reasoning tasks that have a ground-truth answer, whereas off-policy data mostly improves upon more general tasks. We propose SIMPLEMIX : mixing up on- and off-policy data consistently improves performance when using data from a single source.

## Impact Statement

This paper presents work whose goal is to advance the field of Machine Learning. There are many potential societal consequences of our work, none of which we feel must be specifically highlighted here.

## Acknowledgments

This work is supported by ONR grant (N00014-24-1-2089). We sincerely thank Jingyu Zhang, Weiting Tan, Marc Marone, Jeffery Cheng, and Orion Weller for fruitful discussions. We also thank the anonymous reviewers for their helpful suggestions.

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

# Supplemental Material

## A. Hyperparameters

For all of our experiments, we use OpenRLHF (Hu et al., 2024) for training and lm-evaluation-harness (Gao et al., 2024) for evaluation. Initially, we performed hyperparameter sweeps for $\beta = \{0.01, 0.5, 0.1, 1, 5\}$ and max learning rate $= \{5e-7, 1e-6, 5e-6\}$ for initial exploration for DPO. We report the result on $\beta = 0.1$ and max learning rate $= 5e-7$ for **all** of our experiments.

## B. Alpaca Eval 2.0 Prompt Categories

We define the category of prompts as the same as WildChat (Zhao et al., 2024), which contains in total 16 categories. We prompt `gpt-4o-mini` for classifying prompts into one of the 16 categories. We report the distribution of 805 prompts in Alpaca Eval 2.0 (Dubois et al., 2023; 2024) in Figure 7.

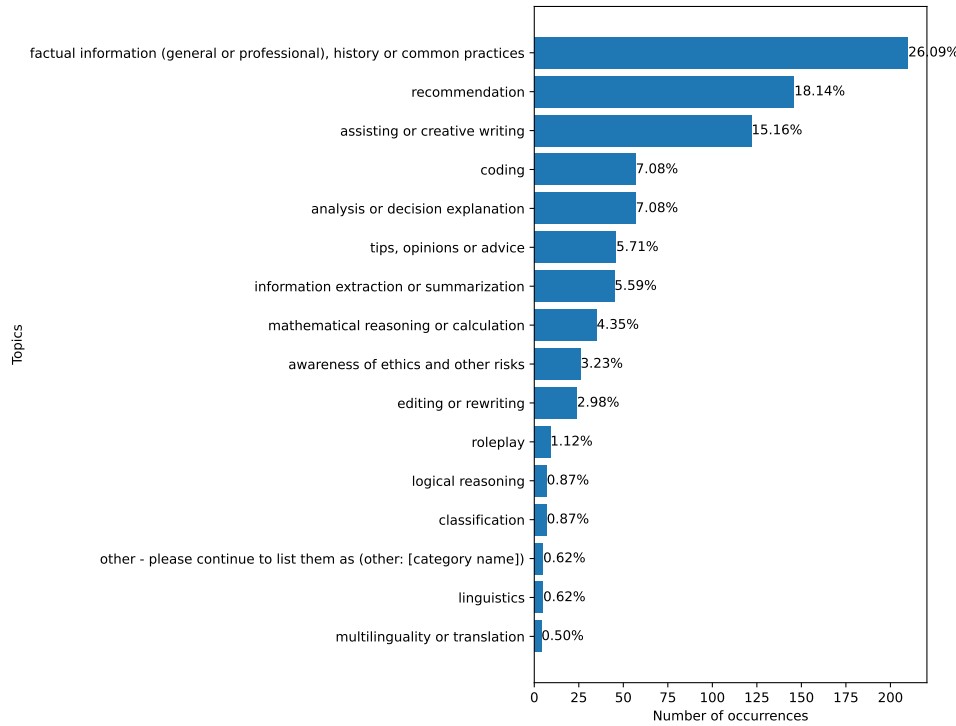

*Figure 7.* Distribution of different categories of prompts in Alpaca Eval 2.0 (Dubois et al., 2023; 2024).

## C. Dataset Details

Here we describe the datasets we used in §4: Ultrafeedback (Cui et al., 2024) and HelpSteer2 (Wang et al., 2024) in detail:

- **Ultrafeedback (Cui et al., 2024)**: **Prompts** included questions in TruthfulQA (Lin et al., 2021), FalseQA (Hu et al., 2023), Evol-Instruct (Xu et al., 2023), UltraChat (Ding et al., 2023), ShareGPT (Chiang et al., 2024b), and the FLAN (Longpre et al., 2023) collection; **Responses** are sampled from a model pool with 17 models: GPT-4, gpt-3.5-turbo, Bard, UltraLM-13B/65B (Ding et al., 2023), WizardLM-7Bv1.1/13B-v1.2/70B-v1.1 (Xu et al., 2023), Vicuna-33B-v1.3 (Ding et al., 2023), LLaMA2-7B/13B/70B-Chat (Touvron et al., 2023), Alpaca-7B (Taori et al., 2023), MPT-30B-Chat (Team, 2023), Falcon-40B-Instruct (Almazrouei et al., 2023), StarChat (Tunstall et al., 2023), and Pythia-12B (Biderman et al., 2023).

- **HelpSteer2 (Wang et al., 2024)**: **Prompts** mainly (over 95%) are sampled from ShareGPT (Chiang et al., 2024b), with a small portion from proprietary user prompts; **Responses** are sampled from Nemotron-2/3/4 (Nvidia et al., 2024),

Mixtral-8x7B-Instruct-v0.1 (Jiang et al., 2023), and human annotators.

## D. Evaluation Details

We use the `lm-eval-harness` (Gao et al., 2024)[4] library for evaluation on all 8 benchmarks (ARC-Challenge, Commonsense QA, Hellaswag, MMLU, OpenQA, PiQA, Winograde, and Ifeval).

The details of the 8 benchmarks are listed below:

- **ARC-Challenge (Clark et al., 2018):** A set of multiple-choice questions focusing on reasoning and scientific knowledge across a range of domains.

- **Commonsense QA (Talmor et al., 2019):** A benchmark designed to evaluate a model's ability to answer questions based on commonsense knowledge, requiring reasoning beyond factual recall.

- **Hellaswag (Zellers et al., 2019):** A benchmark focused on commonsense reasoning and narrative understanding, where models predict the most plausible continuation of an incomplete story.

- **MMLU (Hendrycks et al., 2020):** The Massive Multitask Language Understanding benchmark evaluates models on a diverse set of tasks, including STEM, humanities, and social sciences, measuring performance on human-level tasks.

- **OpenQA (Mihaylov et al., 2018b):** A benchmark for evaluating models on open-domain question answering, testing how well they retrieve and reason over information to answer diverse queries.

- **PiQA (Bisk et al., 2020):** A benchmark focused on physical commonsense reasoning, requiring models to predict the most plausible solution to problems involving physical world interactions.

- **Winograde (Sakaguchi et al., 2020):** A benchmark that assesses models' ability to perform common-sense reasoning in co-reference tasks, where ambiguous pronouns are resolved using contextual clues.

- **Ifeval (Zhou et al., 2023):** A benchmark designed to assess models on their ability to follow precise instructions that are verifiable.

In Alpaca Eval 2.0, for reproducibility, we use greedy decoding with a max length of 2048 tokens.

## E. Full Results

Table 4 and 5 shows the full results on all 8 benchmarks: Commonsense QA (CQA; (Talmor et al., 2019), Hellaswag (HS; (Zellers et al., 2019), Openbook QA (OQA; (Mihaylov et al., 2018a), PIQA (Bisk et al., 2020), Winograde (Sakaguchi et al., 2020), ARC-Challenge (ARC-C; (Clark et al., 2018)), MMLU (Hendrycks et al., 2020), and Ifeval (Zhou et al., 2023), respectively.

## F. Additional Related Works

**Language Model Alignment**  Alignment of language models is typically done at the post-training stage, in order to make the language model prefer certain types of responses. In our paper, we specifically refer the stage **after** supervised fine-tuning (SFT) as to "alignment". In this stage, two paradigms are typically employed: In Reinforcement Learning with Human Feedback (RLHF; (Stiennon et al., 2020; Ouyang et al., 2022)), an on-policy alignment method assigns a reward to on-policy rollouts. Such a reward can be of a separate reward model trained on off-policy generations (Ouyang et al., 2022), hand-crafted heuristics (Bai et al., 2022b), or even AI feedback (Guo et al., 2024). The other paradigm does not require on-policy rollouts and directly manipulates the log probabilities of the offline chosen and rejected responses, hoping that the model generalizes from these offline responses to learn what principle it should follow. Direct Preference Optimization (DPO; (Rafailov et al., 2024)) is the most canonical one that aims to push the log-probs of chosen responses higher and the log-probs of rejected responses lower. Many variants of DPO have been proposed (Pal et al., 2024; Meng et al., 2024; Azar et al., 2024; Xu et al., 2024a) but they mostly used **off-policy** examples. However, the contrastive nature of the DPO loss

---

[4]https://github.com/EleutherAI/lm-evaluation-harness

| | ARC-C | CQA | HSwag | MMLU | OQA | PiQA | WG | Ifeval | Avg. Score |
|---|---|---|---|---|---|---|---|---|---|
| Llama-3.1-8B-Instruct | 51.90 | 77.14 | 59.14 | 67.93 | 33.40 | 79.97 | 73.55 | 44.00 | 60.88 |
| *Experiments on UltraFeedback (Cui et al., 2024)* | | | | | | | | | |
| Off-policy DPO | 53.07 | 77.40 | 59.33 | 68.15 | 34.40 | 80.47 | 74.11 | 46.40 | 61.67 |
| On-policy DPO | 54.18 | 77.81 | 59.88 | 68.60 | 34.80 | 80.41 | 74.98 | 46.03 | 62.09 |
| HyPO (Song et al., 2024b) | 54.86 | 77.64 | 60.09 | 68.20 | **35.40** | 80.63 | 74.59 | 44.60 | 62.00 |
| DPO-Mix-P (Shi et al., 2024) | 53.07 | **77.89** | 59.67 | 68.37 | 34.00 | 80.20 | 74.03 | 45.66 | 61.61 |
| SIMPLEMIX | **55.03** | 77.81 | **61.08** | **69.47** | **35.40** | **81.69** | **75.00** | **48.98** | **63.06** |
| *Experiments on HelpSteer2 (Wang et al., 2024)* | | | | | | | | | |
| Off-policy DPO | 51.79 | 76.99 | 59.22 | 68.11 | 33.60 | 80.09 | 73.88 | **48.61** | 61.54 |
| On-policy DPO | 53.58 | 77.48 | 59.63 | **68.45** | 35.00 | 80.47 | 74.11 | 45.47 | 61.77 |
| HyPO | 53.50 | 76.82 | 59.37 | 68.32 | 34.00 | 80.30 | 73.88 | 47.32 | 61.69 |
| DPO-Mix-P | 53.49 | 77.31 | 59.55 | 68.38 | **34.60** | 79.86 | 74.34 | 44.36 | 61.49 |
| SIMPLEMIX | **54.27** | **78.05** | **59.76** | 68.39 | **34.60** | 80.20 | 74.51 | 47.21 | **62.12** |

*Table 4.* Detailed Benchmark results (ARC Challenge, Common QA, Hellaswag, MMLU, Openbook QA, PiQA, Winograde, Ifeval) on training `Llama-3.1-8B-Instruct` (Dubey et al., 2024) on the Ultrafeedback (Cui et al., 2024) and HelpSteer2 (Wang et al., 2024) dataset.

| Model | ARC-C | CQA | HSwag | MMLU | OQA | PiQA | WG | Ifeval | Avg. Score |
|---|---|---|---|---|---|---|---|---|---|
| Llama-3.1-Tulu-3-8B-SFT | 52.73 | 75.76 | 61.87 | 63.67 | 36.80 | 80.79 | 74.35 | 31.79 | 59.72 |
| *Experiments on UltraFeedback (Cui et al., 2024)* | | | | | | | | | |
| Off-policy DPO | 55.20 | 76.49 | 62.94 | 58.77 | 37.60 | 80.96 | 74.19 | 33.97 | 60.02 |
| On-policy DPO | 54.18 | 76.17 | 62.41 | 58.70 | 37.60 | 80.85 | **74.27** | 33.40 | 59.70 |
| HyPO (Song et al., 2024b) | 55.15 | 76.33 | 61.91 | 62.17 | 37.40 | 80.19 | 73.95 | 33.10 | 60.03 |
| DPO-Mix-P (Shi et al., 2024) | **56.32** | 76.02 | 63.01 | **64.21** | **39.50** | 80.36 | 73.11 | 33.10 | 60.70 |
| SIMPLEMIX | 56.06 | **76.82** | **63.13** | 63.60 | 38.60 | **81.23** | 74.03 | **35.62** | **61.14** |
| *Experiments on HelpSteer2 (Wang et al., 2024)* | | | | | | | | | |
| Off-policy DPO | 52.90 | 76.09 | 61.83 | 63.68 | 37.00 | **80.79** | 73.80 | 33.46 | 59.94 |
| On-policy DPO | 51.90 | 75.92 | 61.88 | 63.77 | 37.00 | 80.74 | 74.03 | 36.41 | 60.21 |
| HyPO (Song et al., 2024b) | 52.65 | **76.33** | 61.85 | **63.90** | 36.40 | 80.63 | **74.43** | 33.46 | 59.96 |
| DPO-Mix-P (Shi et al., 2024) | 52.55 | 76.26 | 62.11 | 63.71 | 36.20 | 80.19 | 74.17 | 35.10 | 60.04 |
| SIMPLEMIX | **52.90** | 76.09 | **63.84** | 63.84 | **37.40** | 80.74 | 74.11 | **40.85** | **61.22** |

*Table 5.* Detailed Benchmark results (ARC Challenge, Common QA, Hellaswag, MMLU, Openbook QA, PiQA, Winograde, Ifeval) on training `Llama-3.1-Tulu-3-8B-SFT` (Lambert et al., 2024a) on the Ultrafeedback (Cui et al., 2024) and HelpSteer2 (Wang et al., 2024).

can also take on-policy examples. Therefore, many on-policy variants of DPO have also been proposed (Xu et al., 2024b; Yuan et al., 2024; Xiong et al., 2024; Guo et al., 2024). Furthermore, there are also off-policy works that are non-contrastive (Ethayarajh et al., 2024). Our paper directly studies the difference (§3) and interaction (§4) of on- vs. off-policy data, disentangling the effect of the contrastive nature of DPO and the non-contrastive nature of standard RLHF.

## G. Per-task Performance of SIMPLEMIX

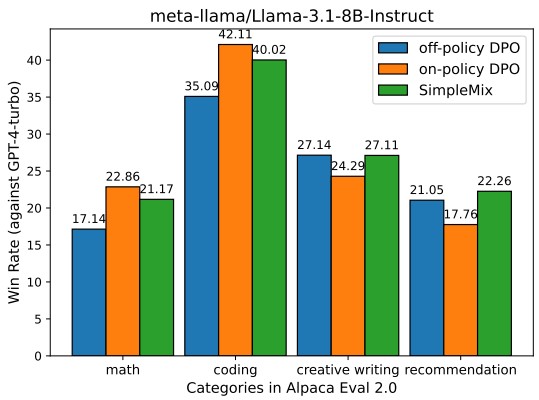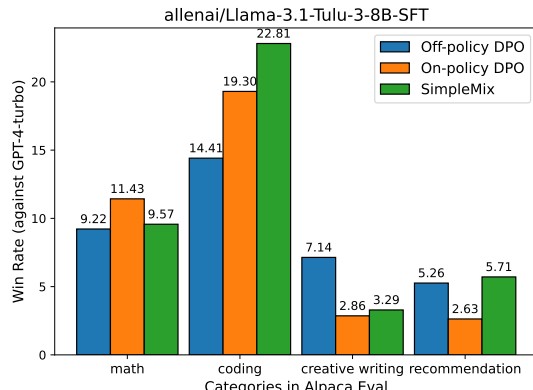

*Figure 8.* Comparison of win rates (against GPT-4-turbo) across different prompt categories in Alpaca Eval 2.0 by training `Llama-3.1-8B-Instruct` and `Tulu-3-8B-SFT` with SIMPLEMIX. SIMPLEMIX offers a good balance between reasoning tasks and open-ended tasks.

## H. Diverse Prompt

Figure 8 shows the prompt we used to elicit the language model to generate a response different from previous responses.

> Please generate another response that is different from all the previous ones.
> It can differ in aspects such as but not limited to tone, formality, strategy, judgment, conciseness, structure. It can also involve asking for clarity if the input prompt is unclear.

*Figure 9.* The prompt used to elicit diverse responses from a language model.

## I. Full Reward Distributions

Figure 10 shows the aggregated histogram of reward of the responses sampled from `Llama-3.1-8B-Instruct` with sampling temperature $\tau = \{0.7, 2, 3\}$ and prompting it to generate diverse responses (Prompting). Figure 11 shows the individual histograms.

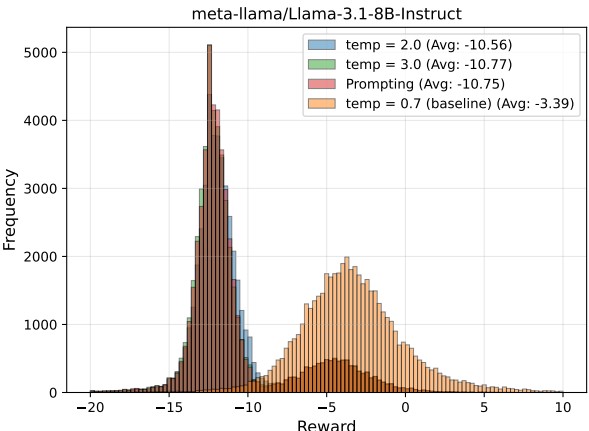

*Figure 10.* Histogram of reward distribution for different sampling temperatures. Increasing the temperature results in generations of lower quality, as measured by our reward model.

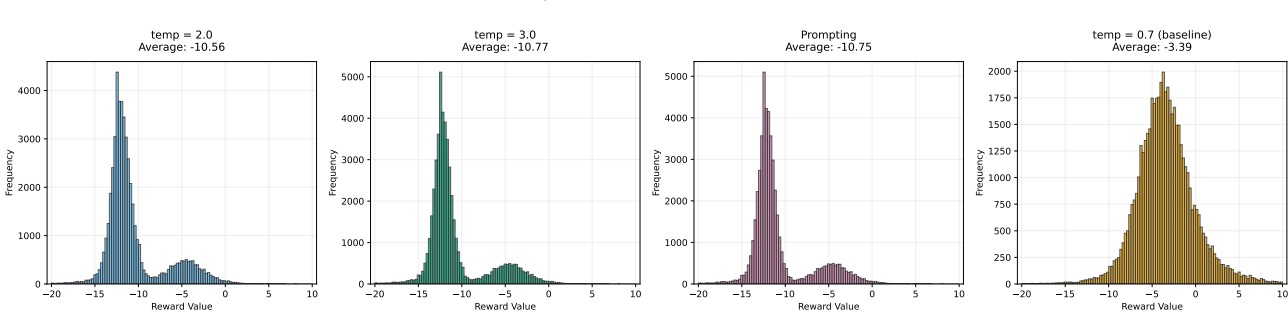

*Figure 11.* Histograms of the reward for sampling from `Llama-3.1-8B-Instruct` with different sampling schemes (prompting & and increasing the temperature).

