# OpenReview forum: "SIMPLEMIX: Frustratingly Simple Mixing of Off- and On-policy Data in Language Model Preference Learning"
_ICML.cc/2025/Conference — ICML 2025 poster_

### Official Review · Reviewer_fktC · 2025-03-07

**Overall Recommendation:** 4

**Summary:**

Authors' first claim that on-policy data are most effective on tasks with objective answers, while off-policy data are most effective on open-ended tasks like creative writing. Analysis on AlpacaEval supports the claim. Then, authors propose a simple method of mixing on-policy and off-policy: just sample from two policies with equal probability: SimpleMix. SimpleMix method makes modest but consistent improvement over strong baselines. Authors also provide good ablations with different mixtures, filtering mechanisms, response diversity.

## update after rebuttal

The new result with DNO further strengthens the benefit of simplicity. I increased my score accordingly.

**Claims And Evidence:**

Authors' first claim is that on-policy data are most effective on tasks with objective answers. This is backed by AlpacaEval 2.0, but it is a bit weak because AlpacaEval has very small samples on Math and Coding. Also, on-policy algorithms are only trained for a single epoch, which could've underestimated their performances. SimpleMix shows consistent benefit over reasonable baseline methods. The improvement is modest, but the consistency makes a convincing case.

**Essential References Not Discussed:**

Methods like Direct Nash Optimization (Rosset et al 2024 https://arxiv.org/abs/2404.03715) do mix samples from online policy and offline policy together, hence SimpleMix is not the only paper which mixes online and offline policy samples. DNO could've been an interesting baseline, at least a non-iterative version.

**Experimental Designs Or Analyses:**

As discussed in Methods And Evaluation Criteria, I checked experimental setups in Section 3.1 and 4.1. Issues are discussed in the section.

In addition, I have a concern that all algorithms are only run for a single epoch. I hypothesize that online policy algorithms benefit more from more epochs since new responses are generated for every epoch. Hence, authors' experiments can be underestimating the benefit of on-policy methods.

**Methods And Evaluation Criteria:**

I checked the experimental setup in Section 3.1. In terms of base model choices, two Llama 3.1-8B based SFT models are good choices. They are performant and represent best practices in the literature. The setup could've been improved with more diverse models, in particular more diverse sizes, although the computational constraint is understandable. AlpacaEval 2.0 is appropriate for this initial experimentation, but its math subset is pretty small (<50) hence I am not sure how much 2% ~ 5% difference in Figure 2 on math is meaningful. Not a major concern because the observation is consistent on Coding, but it could've been more convincing if bigger benchmarks were used for this analysis.

I also checked the experimental setup in Section 4.1. Again base model choices are standard. UltraFeedback is also a good choice as it is well-established and covers various capabilities (for ex per Ivision et al https://arxiv.org/abs/2406.09279). Evaluation on AlpacaEval is good for overall conversation quality, but the rest of benchmarks is a bit too much focused on knowledge tasks, which don't move much from DPO training. I suggest a setup like https://arxiv.org/abs/2406.09279 which more broadly covers coding, safety, truthfulness, etc. Evaluating on coding and reasoning could've helped better validating authors' claim that on-policy performs better on reasoning and off-policy performs better on creative writing. Authors compare against a good set of baselines (HyPO and DPO-Mix-P).

**Other Comments Or Suggestions:**

Summary in line 080-094 is too much of a repeat from only a few paragraphs above. I suggest making the summary more concise so that it becomes less of a repeat.

**Other Strengths And Weaknesses:**

Proposed method is very simple, and hence has a strong potential to be adopted in practice.

**Questions For Authors:**

Paper is clearly written, hence I don't have major questions which would change my evaluation.

**Relation To Broader Scientific Literature:**

The connection to symmetric sampling from Ball et al (2023) is interesting because the method is highly similar, hence establishes a connection to the broader reinforcement learning literature. It could've been nicer if it was discussed more prominently in earlier literature review sections. Currently it is only discussed in Section 5.1.

**Theoretical Claims:**

The paper doesn't make theoretical claims, but make a good connection to theoretical works and compare against them Shi et al 2024, and Song et al 2025.

---

> ### Author Rebuttal · Authors · 2025-04-01
>
> We sincerely thank the reviewer for the insightful comments and suggestions. We appreciate that the reviewer finds our work simple and has a strong potential to be adopted in practice. We are also encouraged that the reviewer finds our paper “clearly written”.
>
> > It could've been nicer if it was discussed more prominently in earlier literature review sections. Currently it is only discussed in Section 5.1.
>
> We thank the reviewer for relating our work to the broader RL literature. We will discuss our relation with existing work in RL that is not about language model alignment earlier in our updated manuscript, and extend our discussion in the appendix.
>
> > Methods like Direct Nash Optimization (Rosset et al 2024 https://arxiv.org/abs/2404.03715) do mix samples from online policy and offline policy together, hence SimpleMix is not the only paper which mixes online and offline policy samples. DNO could've been an interesting baseline, at least a non-iterative version.
>
> We thank the reviewer for bringing up the interesting discussion on DNO. Although the authors of DNO defined it as “a batched on-policy algorithm” (page 9 in [1]), DNO selects the best and worst response from N on-policy generations and one off-policy generation from gpt-4-turbo to perform DPO, therefore it can be seen as an hybrid method. As suggested by the reviewer, we conduct an experiment comparing SimpleMix with one iteration of DNO on Tulu-3.1-8B-SFT, where we sample N = 4 generations and mix it with the off-policy “chosen” generation from Ultrafeedback, and select the best and worst response according to our oracle reward model. We report the results below:
>
>
> ### Alpaca Eval 2.0
>
> | Model                    | LC    | WR    | Std Error | Length |
> |--------------------------|-------|-------|-----------|--------|
> | Tulu 3 SFT + DNO         | 16.22 | 14.15 | 1.29      | 1521   |
> | Tulu 3 SFT + SimpleMix   | 20.64 | 18.02 | 1.36      | 1474   |
>
> Compared to the setting in [1], we changed the number of on-policy generations (5 -> 4), the off-policy model (gpt-4-turbo -> the chosen response in ultrafeedback), and the oracle reward (gpt-4-turbo -> Skywork/Skywork-Reward-Gemma-2-27B)  to make the results comparable to our work. We will make sure to add discussions on DNO in our latest manuscript.
>
> [1] Direct Nash Optimization: Teaching Language Models to Self-Improve with General Preferences (Rosset et al., 2024)

---

> > ### Comment · Reviewer_fktC · 2025-04-02
> >
> > Thanks! I didn't really expect authors to run an experiment for the rebuttal. Thanks for being so open to the suggestion. The new result with DNO further strengthens the benefit of simplicity. I will update my score accordingly.

---

### Official Review · Reviewer_KSRd · 2025-03-13

**Overall Recommendation:** 3

**Summary:**

This paper studies the effect of mixing on-policy and off-policy data when fine-tuning language models using direct preference optimization (DPO). They observe that on-policy data (data generated by the current policy) tends to work better for tasks with clear correct answers, like math and coding, whereas off-policy data (generated by other pre-trained models) is better for open-ended tasks, such as creative writing or recommendations. They propose a simple method called SIMPLEMIX, where on-policy and off-policy data are mixed in equal proportions. Their main findings show SIMPLEMIX consistently improves performance compared to using either data source alone, and it also beats more complicated methods like HyPO and DPO-Mix-P, according to evaluations on AlpacaEval 2.0 and other benchmarks.

**Claims And Evidence:**

The authors claim that mixing on-policy and off-policy data improves alignment performance compared to just using one data source alone. Their main results (Table 2) support this claim, showing SIMPLEMIX outperforms methods like DPO-Mix-P and HyPO. The actual improvement in the length-controlled win rate on AlpacaEval 2.0 is modest (e.g., from around 28 to 30 in Figure 4). Moreover, the authors show that, if off-policy data is curated, the performance of SIMPLEMIX is further improved. They show this on Alpaca Eval 2.0 as well (Figure 6).

**Essential References Not Discussed:**

Nothing noteworthy to me.

**Experimental Designs Or Analyses:**

I checked their experimental designs briefly, especially Figure 4, where they compare various mixing ratios of on-policy and off-policy data. This analysis seems reasonable, clearly showing that a balanced mixture (around 50-50) is slightly better than using either purely on-policy or off-policy data. However, from the results in Figure 4, the performance difference between SIMPLEMIX and the two extremes (purely on/off-policy) is modest. The win-rate accuracy curves are close together, suggesting limited practical impact. Thus, while the analysis itself is sound, the paper doesn’t convincingly demonstrate that the added complexity of mixing datasets is justified by these marginal improvements.

**Methods And Evaluation Criteria:**

The method itself (mixing responses from on-policy and off-policy sources equally) is very straightforward. They evaluate using standard benchmarks like AlpacaEval 2.0 and FineWeb, which makes sense for the alignment task. These benchmarks clearly separate reasoning tasks (math, coding) from subjective tasks (creative writing, recommendations). Overall, their choice of methods and evaluation criteria seems reasonable and appropriate for the problem they’re studying.

**Other Comments Or Suggestions:**

The paper presents a very simple method with clearly explained and straightforward experiments. Overall, the experiments are sound, clearly show modest but consistent improvements, and cover relevant datasets. The main limitation is that the theoretical contribution is minimal—it's mostly applying a known idea (mixing data sources) without significant new insights. The modest size of the observed performance gains also limits its practical significance.

**Other Strengths And Weaknesses:**

Strength:

1. SIMPLEMIX is straightforward and easy to implement.
2. They provide clear experimental results that consistently support their claims.
3. They clarify the conditions (objective vs. subjective tasks) under which on- or off-policy data might be more beneficial.

Weaknesses:

1. The idea itself isn’t very novel—just mixing two data sources is fairly standard.
2.  The reported performance improvements are modest, raising doubts about real-world significance.

**Questions For Authors:**

1. In Figure 6 you show that SIMPLEMIX improves performance much more clearly for Tulu-3 SFT compared to LLaMA-3 Instruct-tuned models. Can you explain why Tulu specifically benefits more from SIMPLEMIX? Is there something specific about Tulu’s training or data that makes mixing data sources especially useful here?

2. In Figure 8, SIMPLEMIX shows very little improvement or sometimes no improvement over the baseline (pure on-policy or off-policy data) when applied to the LLaMA instruct-tuned model across all four task categories (math, coding, creative writing, recommendation). This again relates the Q1 above: it seems that mixing is not useful when the model is already aligned?

Could the authors share some insights on this? Thank you.

**Relation To Broader Scientific Literature:**

They relate their paper clearly to recent debates about using on-policy versus off-policy data in preference alignment, and they discuss recent similar works like HyPO and DPO-Mix-P. The idea of mixing on- and off-policy data isn't new, but clearly showing which tasks benefit from each type of data is a useful contribution. However, their contribution is mostly empirical, with no new theoretical advances.

**Theoretical Claims:**

The paper does not include major new theoretical claims. They mostly rely on intuition and experimental results to justify the SIMPLEMIX idea. There aren't complicated theoretical proofs to verify.

---

> ### Author Rebuttal · Authors · 2025-04-01
>
> We thank the reviewer for the thoughtful review and we would like to extend our gratitude for the reviewer finding our method straightforward and our experiment results clear. We hope to answer the questions and resolve the concerns of the reviewers below:
>
> > The idea itself isn’t very novel—just mixing two data sources is fairly standard.
>
>  - **On its novelty:** Our contribution is not limited to mixing of the two data sources, our work also 1) shows that the two data sources are complimentary (Section 3) - which provides a possible explanation for the performance improvement; 2) conduct detailed ablations (Section 5) on data mixtures and other data curation strategies. **To the best of our knowledge, our work is the first to carefully study the interplay between on- and off-policy data in preference optimization.**
>
>  - **On simplicity:** While we acknowledge that the method is simple, we believe this is our strength. SimpleMix does not introduce additional hyper-parameters. To put it in reviewer `9f8w`’s words, “SIMPLEMIX method is a straightforward but effective way to improve language model alignment without additional computational overhead.”
>
> > The reported performance improvements are modest, raising doubts about real-world significance.
>
>  - **Improvements**: While the reported performance improvements may appear modest, they are averaged across **eight** benchmarks. As reviewer `fktC` denotes “The improvement is modest, but the consistency makes a convincing case.”
>  - **Real-world significance** Tulu 3 [1] is a concurrent effort that is similar to our setting and demonstrates a 3-point average improvement across benchmarks. This is a meaningful gain, especially considering that mixing the two data sources incurs minimal cost. As Reviewer `fktC` denotes, our method “has a strong potential to be adopted in practice.”
>
> > Can you explain why Tulu specifically benefits more from SIMPLEMIX? Is there something specific about Tulu’s training or data that makes mixing data sources especially useful here?
> > In Figure 8, SIMPLEMIX shows very little improvement or sometimes no improvement over the baseline (pure on-policy or off-policy data) when applied to the LLaMA instruct-tuned model across all four task categories (math, coding, creative writing, recommendation). This again relates the Q1 above: it seems that mixing is not useful when the model is already aligned?
>
> We aggregated the two questions here: we conjecture that Tulu-8B-SFT have not gone through a preference optimization process while Llama-3.1-8B-Instruct have already gone through extensive SFT and DPO training, thus making it easier to improve Tulu’s performance compared to improving Llama-3.1-8B-Instruct’s performance.
>
> References:
> [1] Tulu 3: Pushing Frontiers in Open Language Model Post-Training (Lambert et al., 2024)

---

### Official Review · Reviewer_9f8w · 2025-03-15

**Overall Recommendation:** 2

**Summary:**

The paper investigates the interplay between on-policy and off-policy preference data in aligning large language models (LLMs) with human preferences. It presents the key finding that on-policy data is more effective for reasoning tasks (e.g., math, coding), whereas off-policy data performs better in open-ended tasks (e.g., creative writing, recommendations). Based on this observation, the authors propose SIMPLEMIX, a method that combines both data sources in a straightforward manner.

**Claims And Evidence:**

Most claims are well-supported.

**Essential References Not Discussed:**

Lacks discussion on recent hybrid RL preference optimization and adaptive weighting.

**Experimental Designs Or Analyses:**

Yes, but it lacks human evaluation.

**Methods And Evaluation Criteria:**

Yes, but relies heavily on LLM-based evaluation; lacks human validation.

**Other Comments Or Suggestions:**

See Strengths And Weaknesses Part.

**Other Strengths And Weaknesses:**

**Strengths:**

1. The proposed SIMPLEMIX method is a straightforward but effective way to improve language model alignment without additional computational overhead.

2. The results are statistically significant and demonstrate clear trends in performance across different tasks.

**Weaknesses:**

1. The paper lacks a theoretical foundation to explain why SIMPLEMIX works better than other hybrid approaches.

2. Benchmarks like Alpaca Eval 2.0 and Ifeval involve LLM-based evaluation, which can be biased or manipulated. Including more human evaluation results (e.g., actual user studies) would add credibility to the findings.


3. While the study identifies task-dependent benefits of on- and off-policy data, it does not explore task-specific weighting or adaptation. A potential extension could involve dynamically adjusting the data mixture ratio based on task type.

**Questions For Authors:**

See Strengths And Weaknesses Part.

**Relation To Broader Scientific Literature:**

Builds on preference learning, hybrid RL, and DPO.

**Theoretical Claims:**

No formal proofs provided; lacks theoretical justification for SIMPLEMIX effectiveness.

---

> ### Author Rebuttal · Authors · 2025-04-01
>
> We sincerely thank the reviewer for the thoughtful comments and suggestions. We are thankful that the reviewer finds our method “straightforward but effective” and our results “demonstrate clear trends”. We hope to resolve the concerns below:
>
> > Benchmarks like Alpaca Eval 2.0 and Ifeval involve LLM-based evaluation, which can be biased or manipulated. Including more human evaluation results (e.g., actual user studies) would add credibility to the findings.
>
> It is possible there may be some misunderstanding of the scope of our evaluation. In our work, we have chosen to report scores of **nine** benchmarks, only one of them is LLM-based evaluation (Alpaca Eval 2.0). The other 8 benchmarks include but are not limited to world knowledge acquisition (MMLU), commonsense reasoning (Hellaswag), precise instruction following (IFEval), open-domain question answering (OpenQA). We believe that this is a practical and comprehensive setting since the same benchmarks are also adopted by FineWeb [6] in pre-training data curation. Detailed descriptions of our evaluation benchmarks can be found at Appendix D.
> We acknowledge that adding real user studies would improve the credibility of our work. Since the cost of high quality human annotators is prohibitively large, we opted to use Alpaca Eval 2.0 due to it having a 0.95 correlation with real human annotators while only having 0.8% of the cost according to [7].
>
> > The paper lacks a theoretical foundation to explain why SIMPLEMIX works better than other hybrid approaches.
>
> There exists a mismatch between theory of existing works and practice in LM alignment, and we conjecture that might be the reason why SIMPLEMIX works better:
>
> - HyPO [1] proposes to perform an off-policy DPO while using on-policy data to minimize the KL divergence between the current policy ($\pi_\theta$) and the reference policy ($\pi_\text{SFT}$). The motivating assumption for adding the additional KL regularization is that the “DPO implicit reward” is unbounded and can lead to infinite KL. In practice, the literature has witnessed the *opposite* trend [2, 3, 4] where KL regularization might not be necessary for LM alignment (because of the reference model’s unreliability, leading to unreliable KL values). Our work combines on- and off-policy data *without explicit regularization* of the KL compared to HyPO, removing the strong KL regularization in HyPO might be a reason that SimpleMix works better.
>
> - DPO-Mix-P [5] samples from an interpolation between $\pi_\theta$ (the current policy) and $\pi_\text{SFT}$ (reference policy). The theoretical foundation of DPO-Mix-P is designed for faster convergence in terms of DPO loss (achieving a lower DPO loss for less iterations). In practice, the recent works have shown that lower DPO loss doesn’t always go hand-in-hand with better alignment (“Alignment Gap” [6]). Therefore, achieving a lower DPO loss, or equivalently, a higher ranking accuracy, might not contribute to a better aligned model.
>
> > While the study identifies task-dependent benefits of on- and off-policy data, it does not explore task-specific weighting or adaptation. A potential extension could involve dynamically adjusting the data mixture ratio based on task type.
>
> In section 3, we have shown the complementariness of on- and off-policy data on carefully selected subtopics in Alpaca Eval 2.0. However, in reality, user prompts rarely fall into a single category as many queries require both understanding the user’s personal preference, but also following verifiable constraints. For example, the query from UltraFeedback “explain cache keys to me like im 5” requires both objective knowledge about cache keys and adjustments of tone and style. Therefore, we decided to be safe and only combine two sources of data to achieve a balanced performance across all types of tasks. We agree with the reviewer that algorithmic adjustment of the sampling method (on- or off-policy) based on the prompt category would be a promising future direction.
>
> References:
>
> [1] The Importance of Online Data: Understanding Preference Fine-tuning via Coverage (Song et al., NeurIPS 2024)
>
> [2] Contrastive Preference Optimization: Pushing the Boundaries of LLM Performance in Machine Translation (Xu et al., ICML 2024)
>
> [3] SimPO: Simple Preference Optimization with a Reference-Free Reward (Meng et al. NeurIPS 2024)
>
> [4] DAPO: An Open-Source LLM Reinforcement Learning System at Scale (Yu et al., 2025)
>
> [5] The Crucial Role of Samplers in Online Direct Preference Optimization (Shi et al., ICLR 2025)
>
> [6] Preference Learning Algorithms Do Not Learn Preference Rankings (Chen et al., NeurIPS 2024)
>
> [7] FineWeb: decanting the web for the finest text data at scale (Penedo et al., 2024)
>
> [8] MixEval: Deriving Wisdom of the Crowd from LLM Benchmark Mixtures (Ni et al., NeurIPS 2024)

---

### Decision · Program_Chairs · 2025-05-01

**Decision:**

Accept (poster)

**Comment:**

My recommendation is to accept the paper.

The paper proposes a simple preference alignment approach that mixes on-policy and off-policy in equal parts in a DPO objective. The method follows from existing discussion in the literature about the relative strengths of on- and off-policy data, and significantly simplifies existing approaches that attempt to mix data from these distributions. The method shows significant improvement over standard baselines, and consistent (if small) improvement over more complex but "mixed" baselines.

There was strong disagreement among reviewers about this paper. The key disagreements came down to (1) the necessity of human evaluation, and (2) necessity for some theoretical justification.

One reviewer made a strong case that human evaluation is important for the class of tasks that are more subjective, while another noted that much progress has been made on preference alignment (even on subjective criteria) using automatic evaluation. While I am sympathetic to the idea that human evaluation is ultimately necessary, I think the primary object of study here is the optimization algorithm itself, so I think it is valid to perform experiments with a proxy, although human evaluation or replication on multiple diverse proxy reward models could make the case more strongly.

Regarding theoretical justification, I agree that it would be useful for the literature to have more theoretical threads to pull on here, but I think that the main contribution of this work is to show that something simple and easily implemented does work. The algorithm itself is immediately useful, and there is room for subsequent work to take this observation and work out the science.

I would encourage the authors to address these issues directly in the text of the paper.